# Dispersal history and bidirectional human-fish host switching of invasive, hypervirulent *Streptococcus agalactiae* sequence type 283

**Daniel Schar**[1]*, **Zhenyu Zhang**[2], **Joao Pires**[3], **Bram Vrancken**[1,4], **Marc A. Suchard**[2,5], **Philippe Lemey**[4], **Margaret Ip**[6], **Marius Gilbert**[1,7], **Thomas Van Boeckel**[3,8], **Simon Dellicour**[1,4]

**1** Spatial Epidemiology Laboratory, Université Libre de Bruxelles, Brussels, Belgium, **2** Department of Biostatistics, Fielding School of Public Health, University of California, Los Angeles, Los Angeles, CA, United States of America, **3** Institute for Environmental Decisions, ETH Zurich, Zurich, Switzerland, **4** Department of Microbiology, Immunology and Transplantation, Rega Institute for Medical Research, KU Leuven, Leuven, Belgium, **5** Department of Human Genetics, David Geffen School of Medicine at UCLA, University of California, Los Angeles, Los Angeles, CA, United States of America, **6** Department of Microbiology, Faculty of Medicine, Prince of Wales Hospital, The Chinese University of Hong Kong, Hong Kong SAR, China, **7** Fonds National de la Recherche Scientifique, Brussels, Belgium, **8** Center for Diseases Dynamics, Economics, and Policy, New Delhi, India

* dlschar@gmail.com, Daniel.Schar@ulb.be

**Data Availability Statement:** All data generated from this study and R code used to analyze the

## Abstract

Human group B *Streptococcus* (GBS) infections attributable to an invasive, hypervirulent sequence type (ST) 283 have been associated with freshwater fish consumption in Asia. The origin, geographic dispersion pathways and host transitions of GBS ST283 remain unresolved. We gather 328 ST283 isolate whole-genome sequences collected from humans and fish between 1998 and 2021, representing eleven countries across four continents. We apply Bayesian phylogeographic analyses to reconstruct the dispersal history of ST283 and combine ST283 phylogenies with genetic markers and host association to investigate host switching and the gain and loss of antimicrobial resistance and virulence factor genes. Initial dispersal within Asia followed ST283 emergence in the early 1980s, with Singapore, Thailand and Hong Kong observed as early transmission hubs. Subsequent intercontinental dispersal originating from Vietnam began in the decade commencing 2001, demonstrating ST283 holds potential to expand geographically. Furthermore, we observe bidirectional host switching, with the detection of more frequent human-to-fish than fish-to-human transitions, suggesting that sound wastewater management, hygiene and sanitation may help to interrupt chains of transmission between hosts. We also show that antimicrobial resistance and virulence factor genes were lost more frequently than gained across the evolutionary history of ST283. Our findings highlight the need for enhanced surveillance, clinical awareness, and targeted risk mitigation to limit transmission and reduce the impact of an emerging pathogen associated with a high-growth aquaculture industry.

data are available on the Zenodo public repository (10.5281/zenodo.8345450).

**Funding:** T.V.B. was supported by the Swiss National Science Foundation and the Branco Weiss Foundation. The funders had no role in study design, data collection and analysis, decision to publish, or preparation of the manuscript.

**Competing interests:** The authors have declared that no competing interests exist.

## Introduction

Group B *Streptococcus* (GBS; *Streptococcus agalactiae*) is carried in the gastrointestinal and urogenital tracts and is well recognized as a cause of neonatal sepsis and meningitis as well as severe disease in pregnant adults and the immunocompromised [1]. Recent outbreaks of an invasive, hypervirulent GBS sequence type (ST) 283 responsible for severe disease in younger adults with fewer comorbidities have been associated with handling and consumption of freshwater fish [2–7]. Clinically, severe disease attributable to ST283 infection is characterized by sepsis, septic arthritis, meningitis, and infective endocarditis [7].

Retrospectively identified ST283 human infections were reported from Hong Kong from samples collected in 1993 and were exclusively associated with invasive sites in non-pregnant adults [7]. A 2015 Singapore epidemic of invasive GBS infections identified co-incident clonal ST283 isolates from human case-patients and fish, complementing a case-control study associating raw fish consumption with ST283 infection, and confirming this strain is transmissible as a freshwater fish foodborne disease [3, 4]. Outside of Asia, ST283 has also been reported sporadically from isolates collected in the United States, the United Kingdom, and from osteoarticular infections reported in France [8].

In aquaculture, GBS is responsible for substantial fish mortality and production loss [9], and both ST283 and a single locus variant (ST491) have been reported from diseased fish in southeast Asia [10, 11]. Isolation of ST283 in 2016 from fish in Brazil indicates this strain may currently be undergoing further global dissemination [12].

The foodborne transmission identified in Singapore as a conduit for ST283 invasive human disease suggests that freshwater fish consumption and contact may be under-appreciated as the source of at least some severe, invasive GBS disease globally [5, 13, 14]. Furthermore, whereas predominantly *tet*(M) gene carriage encoding tetracycline resistance is frequently reported in human-adapted GBS isolates [15], heterogeneous rates of tetracycline resistance gene carriage have been reported in ST283 from fish and human isolates, with putative loss of tetracycline resistance gene events [5, 16]. An enhanced understanding of the origin of ST283, its international distribution, host transitions and the acquisition and loss of antimicrobial resistance and virulence genes holds potential to inform targeted risk mitigation reducing morbidity and mortality associated with this emerging pathogen [13].

Here, we analyze the sequences of 328 genomes from ST283 isolates collected between 1998 and 2021 from eleven countries across four continents. We performed Bayesian phylodynamic and phylogeographic analyses to reconstruct the evolutionary and dispersal history of ST283. We further investigate host switching events and the gain and loss of antimicrobial resistance and virulence factor genes. These approaches provide insight into the evolutionary history of ST283 to inform surveillance and interventions for a pathogen closely associated with freshwater aquaculture as that industry experiences continued global growth.

## Results

### Time scale for the emergence of ST283

Root-to-tip distance was correlated with isolation date (Fig A in S1 Text) to assess the presence of a temporal signal within the genomic dataset. The time to the most recent common ancestor of ST283 inferred from the time-calibrated phylogeny was estimated as 1982 (95% highest posterior density (HPD) interval: 1976 to 1987; Fig 1 and Fig B in S1 Text).

### Geographic expansion outside of Asia is recent

Heterogeneous sampling effort among locations inherent to our dataset precluded drawing any conclusion regarding the root state location. This limitation is confirmed by an analysis in

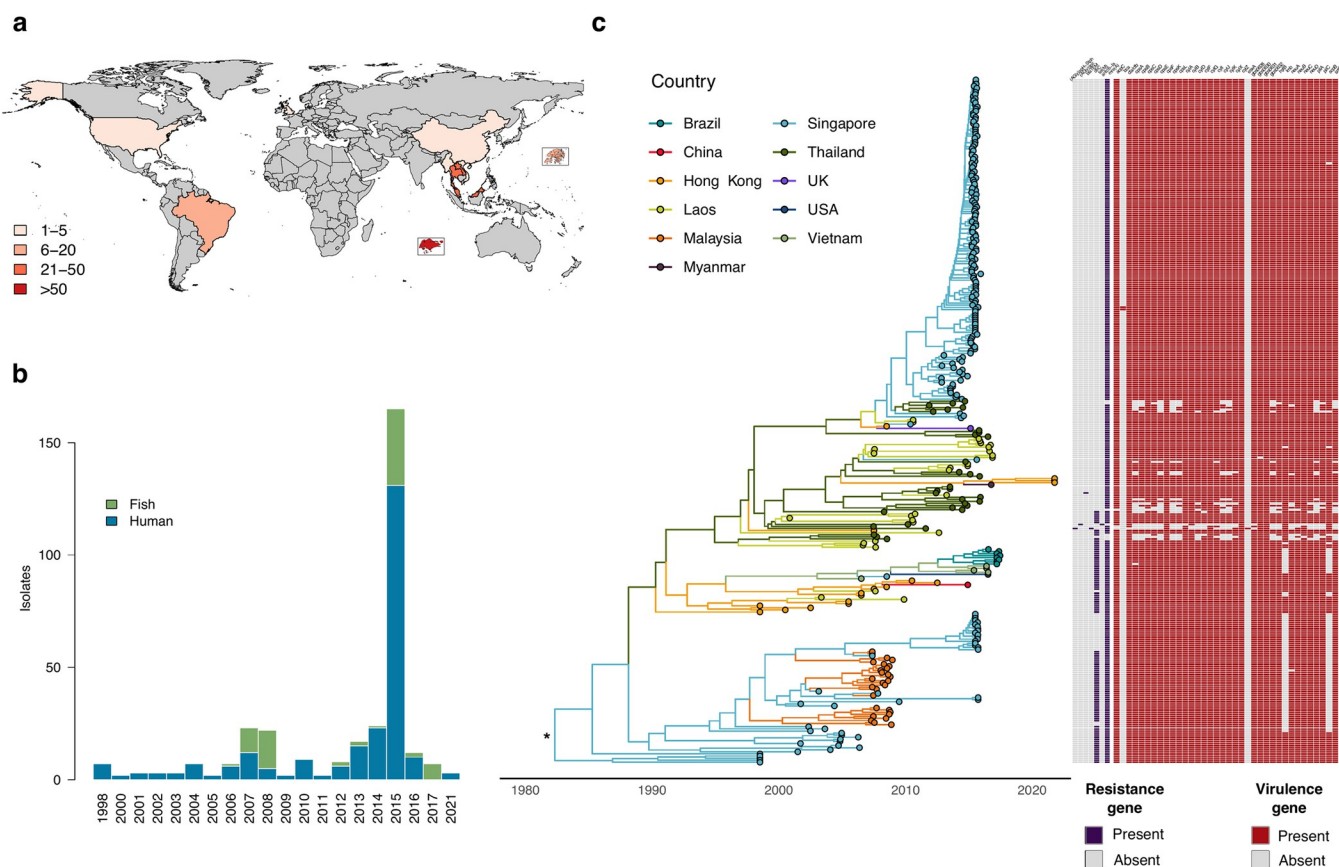

**Fig 1. Spatio-temporal distribution of GBS ST283 isolates and inferred time-scaled phylogenetic tree. (a)** Map of sampling location for ST283 isolates in the study (*n* = 328). Insets are Singapore (bottom) and Hong Kong (right). **(b)** Distribution of isolate sampling dates and host origin. **(c)** Annotated maximum clade credibility (MCC) tree resulting from a discrete phylogeographic analysis. Tip nodes and branches are colored according to the country of origin and the country inferred at ancestral nodes. Asterisk (*) indicates that, after having taken the heterogeneity of the sampling effort among sampled locations into account, no meaningful support was identified for ancestral root node location (see the Results section for further detail). A table of antimicrobial resistance (blue) and virulence factor (red) gene presence or absence is displayed for each isolate in the study. The base layer of the map is available at https://www.naturalearthdata.com/downloads/10m-cultural-vectors/10m-admin-0-countries/.

which the location states are randomly swapped among the tips of the tree during the phylogeographic reconstruction, which yielded a posterior probability for Singapore as the tree root location (*p* = 0.80) very similar to the posterior probability obtained through the standard phylogeographic analysis (*p* = 0.91; Table B in S1 Text). This result indicates that the finding of Singapore at the ancestral root node is not informed by the phylogenetic information but almost exclusively by the oversampling of that particular location.

Discrete phylogeographic analysis identified a posterior mean of 35 (95% HPD interval: 33 to 39) independent transition events between countries across the evolutionary history of ST283. The discrete phylogeographic reconstruction was analyzed in decadal bins, beginning in 1981 and continuing through the most recent sample collection in October 2021 (Fig 2). Following emergence, ST283 experienced an initial period of intracontinental expansion within Asia. Multiple transition events were inferred from Thailand to Hong Kong and Laos; Singapore to Malaysia; and from Hong Kong to Laos and Vietnam between the date of emergence and 2000, continuing through 2010. Thailand, Hong Kong, and Singapore appeared as central hubs for early dissemination of ST283, accounting for 62.1%, 21.5%, and 16.4% respectively of all supported transition events between 1981 and 2001. The earliest supported

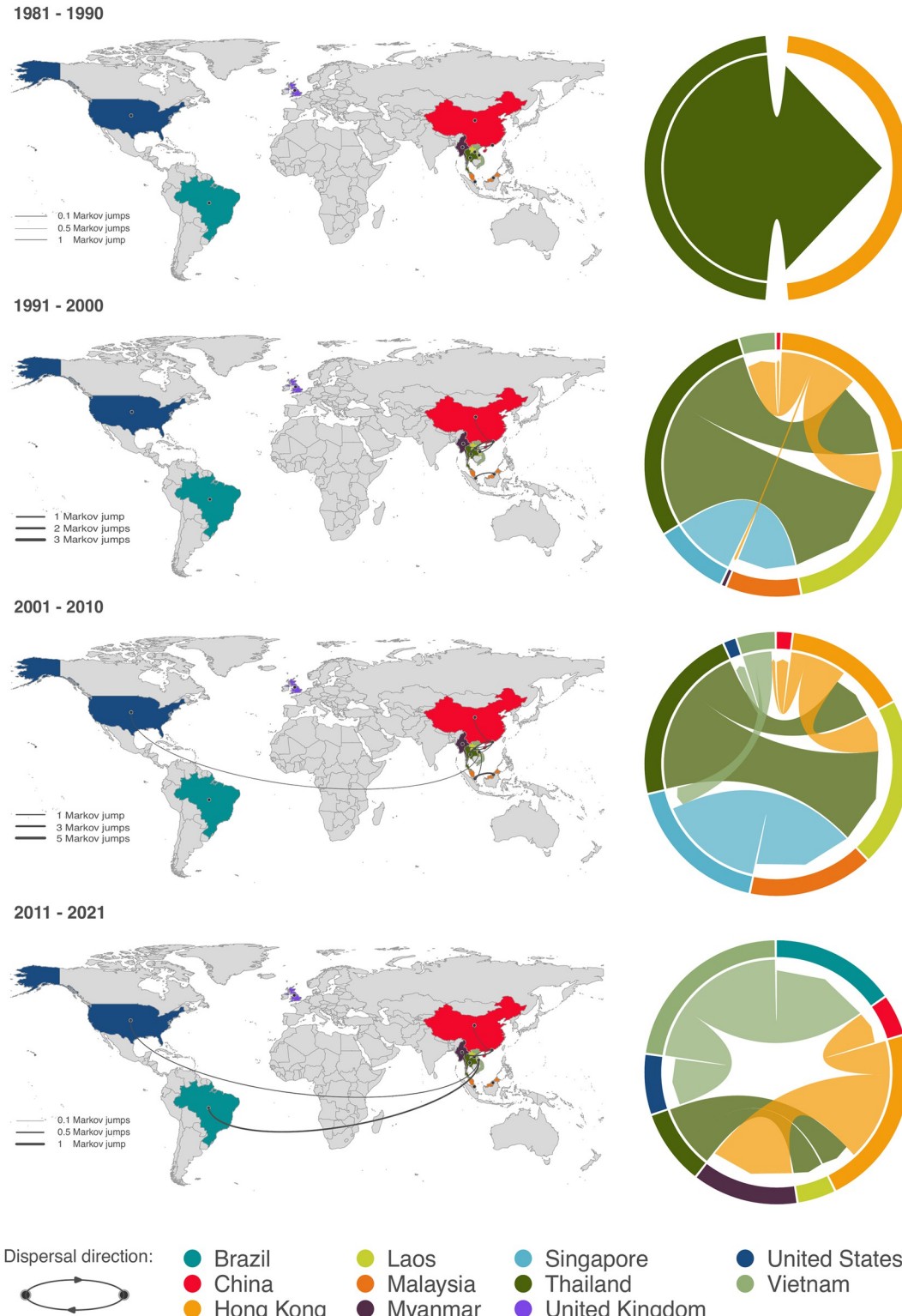

**Fig 2. Discrete phylogeographic analysis of the dispersal history of GBS ST283 between 1981 and 2021.** Intracontinental and intercontinental transition events are inferred as Markov jumps. Maps display transition events by decade and are accompanied by circular migration flow plots, in which transitions out of a country are represented by arrows originating at the outer ring and ending in an arrowhead offset from the destination country. Arrow width is proportional to the magnitude of the Markov jumps. Only transition events associated with an adjusted Bayes factor support > 20 are displayed, a threshold

value corresponding to strong statistical support (see Methods for further detail). The base layer of the map is available at https://www.naturalearthdata.com/downloads/10m-cultural-vectors/10m-admin-0-countries/.

intercontinental transition was inferred between 2001 and 2011 from Vietnam to the United States. In the most recent decade (2011–2021), Hong Kong and Thailand continued to seed international dissemination (34.3% and 18.7% of supported transition events, respectively), and Vietnam remained as a source of intercontinental ST283 movement. The rate of international dispersal of ST283 increased between 1996 and 2010 (Fig C in S1 Text).

## ST283 exhibits frequent bidirectional host switching

Heterogeneous host sampling distribution in the dataset precluded inference of supported Markov jumps between hosts: fish-to-human and human-to-fish transition events were associated with an adjusted Bayes factor ($BF_{adj}$) support equal to and smaller than 1, respectively (see the Methods section for further detail on how Bayes factor supports were computed while considering heterogeneous sampling effort among host types). To account for this sampling bias, ancestral host state transitions were inferred by maximum likelihood estimation on downsampled phylogenies (see the Methods section for further detail). More human-to-fish transitions (median = 9; IQR: 8 to 9) were observed than fish-to-human (median = 2; IQR: 1 to 3) transitions across the evolutionary history of ST283 (Fig 3). This trend was consistent when controlling for the year of sampling as well as the phylogenetic diversity associated with each host type. Total human-to-fish transitions were greater than total fish-to-human transitions in each of 1,000 downsampled phylogenies.

## Multi-drug resistance gene carriage is low in ST283

Seven antimicrobial resistance genes (ARGs) were identified, associated with phenotypic resistance to five classes of antibiotics: aminoglycosides, beta-lactams, dihydrofolate reductase inhibitors (trimethoprim), macrolides, and tetracyclines. Across all isolates, the *mre*(A) gene (321 isolates, 97.9%) and *tet*(M) gene (96 isolates, 29.3%) were most frequently carried, conferring resistance when expressed to macrolides and tetracyclines, respectively; all other ARGs were identified in single instances each. One isolate collected from a human bacteremic patient in Thailand in 2011 carried three ARGs—(AGly)*apH-Stph*, *dfrC*, and *mre*(A)—and was the only isolate in the dataset carrying the *esxA* gene, recently identified as encoding a pore-forming protein important in GBS pathogenesis [17].

## Antimicrobial resistance and virulence genes are more frequently lost than gained across the evolutionary history of ST283

Discrete trait analyses of antimicrobial resistance genes revealed that the *tet*(M) and *mre*(A) genes were lost more frequently than they were gained across the ST283 evolutionary history. Similarly, GBS virulence genes associated with human adaptation—*hylB* (bacterial invasion and dissemination from initial site of infection), *lmb* (adherence), and *scpB* (neutrophil recruitment inhibition)—were lost more frequently than gained (Table 1 and Fig 4).

Trait correlations across the evolutionary history of ST283 were captured by applying a phylogenetic multivariate probit model capable of identifying conditional dependencies amongst any two traits after removing the effects of other traits–the so-called partial correlations. We identify a positive partial correlation between isolate origin from a human host and the *tet*(M) gene encoding tetracycline resistance (posterior median = 0.43). Regarding virulence factor genes, positive partial correlations were identified between human host and the *lmb* (posterior median = 0.25) and *cpsA* (posterior median = 0.31) genes (Fig 5). A partial

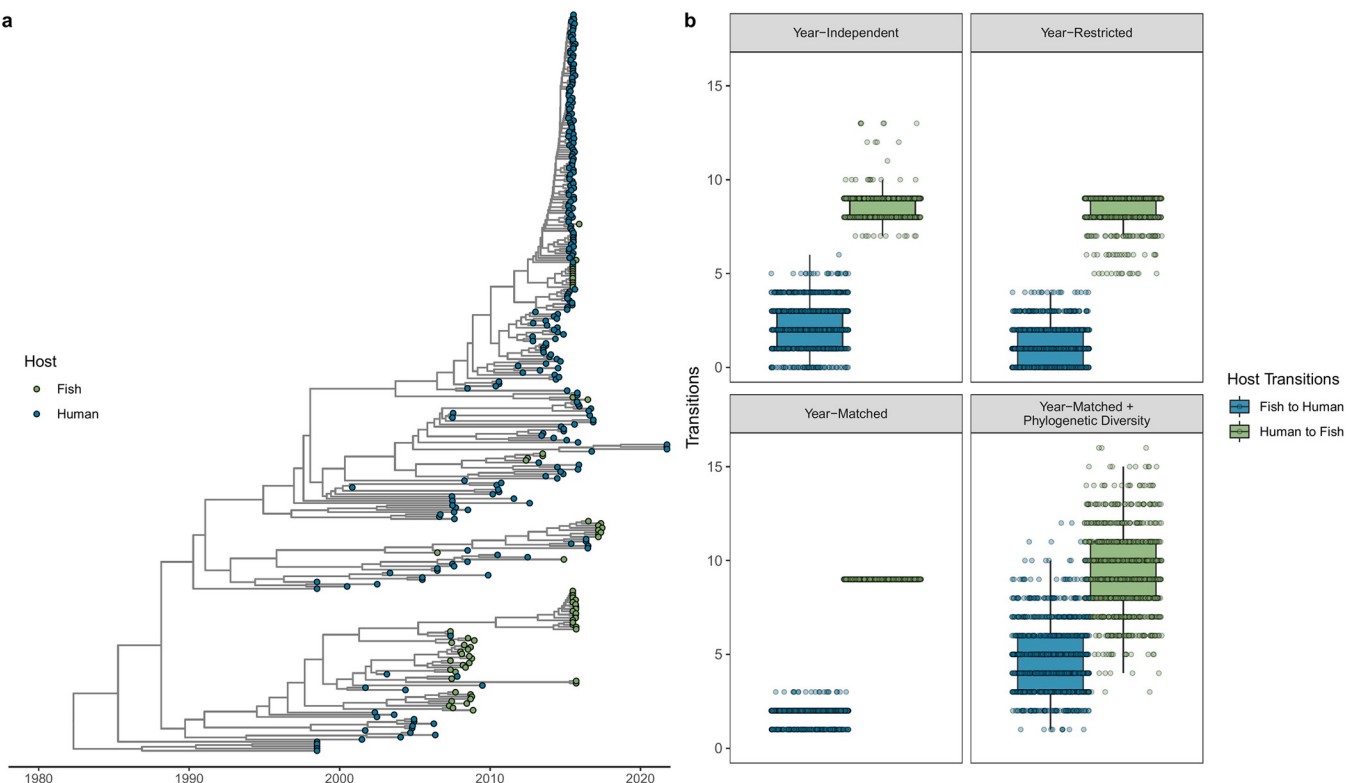

**Fig 3. Bidirectional host switching in GBS ST283. (a)** Host taxa (human or fish) are displayed in the time-scaled maximum clade credibility (MCC) tree resulting from the Bayesian phylogenetic analysis. **(b)** As inference of Markov jumps between hosts were not supported using the full dataset (see the text for further detail), ancestral host state transitions (human-to-fish and fish-to-human) were inferred by maximum likelihood estimate on 1,000 replicate downsampled phylogenies (see Methods for further detail). First, the full dataset was downsampled to produce equal numbers of human and fish origin isolates in the resulting phylogenies without any constraint on sampling year ('year-independent'; n = 154). To compare the effect of sampling year on this downsampling procedure, we conducted two year-constrained analyses: in the first, we worked with a downsampled dataset consisting of equal numbers of isolates originating from human and fish hosts using only isolates within a window of years where both human and fish host isolates were represented ('year-restricted'; n = 140). In the second analysis, we worked with the downsampled dataset from the year-restricted analysis, but further year-matched isolates to include equal numbers of human and fish origin isolates in each year as determined by the minimum number available for either host in that year ('year-matched'; n = 116). Finally, to account for uneven phylogenetic diversity associated with host type, a third analysis is conducted in which monophyletic clusters of sequences collected from the same host type are first subsampled to be represented by a single sequence prior to further subsampling as described in the 'year-matched' analysis (('year-matched + phylogenetic diversity'; n = 32). The horizontal box lines represent the first quartile, the median, and the third quartile. Whiskers denote the range of points within the first quartile −1.5 × the interquartile range and the third quartile +1.5 × the interquartile range. Ancestral host state transitions inferred in each of 1,000 replicate downsampled phylogenies are represented in their respective downsampling approach panel by a dot with horizontal jitter for visibility.

positive correlation was identified between fish host and the *gbs0632* gene (posterior median = 0.24). Other virulence factor gene trait correlations were considered as interactions governing gene function and expression. A positive partial correlation was identified between *lmb* and *scpB* (posterior median = 0.90), virulence factor genes involved in host cell adhesion and prevention of neutrophil recruitment, respectively. Positive partial correlation was identified between *cpsL*—a component of the capsular protein *cps* operon involved in molecular mimicry and immune evasion [18]—and *gbs0632*, which contributes to the GBS binding pilus architecture (posterior median = 0.61).

## Discussion

Our findings confirm the emergence of GBS ST283 in Asia in the latter half of the twentieth century, corresponding with the start of expansive growth in freshwater aquaculture

**Table 1. Antimicrobial resistance and virulence gene gains and losses.** Median transitions from absence to presence (gained) and presence to absence (lost) with their 95% highest posterior density (HPD) intervals. Transitions are calculated from the discrete trait analysis post-burn-in posterior distribution implemented in BEAST.

| | | 95% HPD Interval | |
| --- | --- | --- | --- |
| Gene | Median | Lower | Upper |
| *tet*(M) | | | |
| Gained | 2 | 0 | 3.23 |
| Lost | 5 | 4.20 | 9.54 |
| *mre*(A) | | | |
| Gained | 0 | 0 | 0.07 |
| Lost | 4 | 3.95 | 4.05 |
| *hylB* | | | |
| Gained | 0 | 0 | 3.07 |
| Lost | 9 | 5.89 | 9.42 |
| *lmb* | | | |
| Gained | 0 | 0 | 1.05 |
| Lost | 5 | 4.95 | 5.05 |
| *scpB* | | | |
| Gained | 0 | 0 | 1.07 |
| Lost | 8 | 7.94 | 8.06 |

production driven primarily by Asia [5, 19]. Limited availability of ST283 genomes and a dataset characterized by heterogeneous sampling effort could however influence our findings. Future work to identify and sequence ST283 isolates from under-represented regions may elucidate missing diversity and transition events amongst locations. The time-calibrated phylogeny in this study infers a most recent common ancestor (MRCA) for ST283 in 1982, which aligns closely with a previous estimate of emergence in 1985 [5]. Our analysis yields an emergence date that precedes by more than a decade a MRCA estimate (1994) obtained by evaluating isolates sampled exclusively from humans [16]. The observation that isolates from humans and fish cluster together across multiple clades within the tree suggests that the difference in MRCA estimates is not due solely to the host origin. The short window measuring approximately a decade between the first reported human cases in 1993 and date of ancestral origin indicates that ST283 may have emerged from a proximal GBS lineage having nearly acquired capacity for–or being fully capable of–producing severe clinical disease in humans, although missing genetic diversity may affect MRCA estimates.

Evidence presented in this study suggests that ST283 may be following an expanding geographic range trajectory outside of Asia. The frequency of intercontinental transitions (Markov jumps from or to Asia) increased between 2006 and 2015 before declining between 2016 and 2021 (Fig 2 and Fig C in S1 Text). Whether this trend reflects an actual decline in intercontinental movement over the last five years or rather is influenced by comparatively fewer isolates in the dataset after 2016 ($n = 22$, 6.7%), and particularly from 2018 to 2021 ($n = 3$, 0.91%), will require further verification.

Laos, Hong Kong, and Singapore isolates are represented across clades, suggesting multiple introduction events, consistent with our discrete phylogeographic analysis showing active dispersal histories within Asia beginning in 1991 and accelerating into the decade from 2001 to 2010.

We observe more frequent human-to-fish than fish-to-human host switching–a trend that remains consistent when restricting the analysis to include the same number of sampled hosts in each year and when controlling for differences in phylogenetic diversity associated with

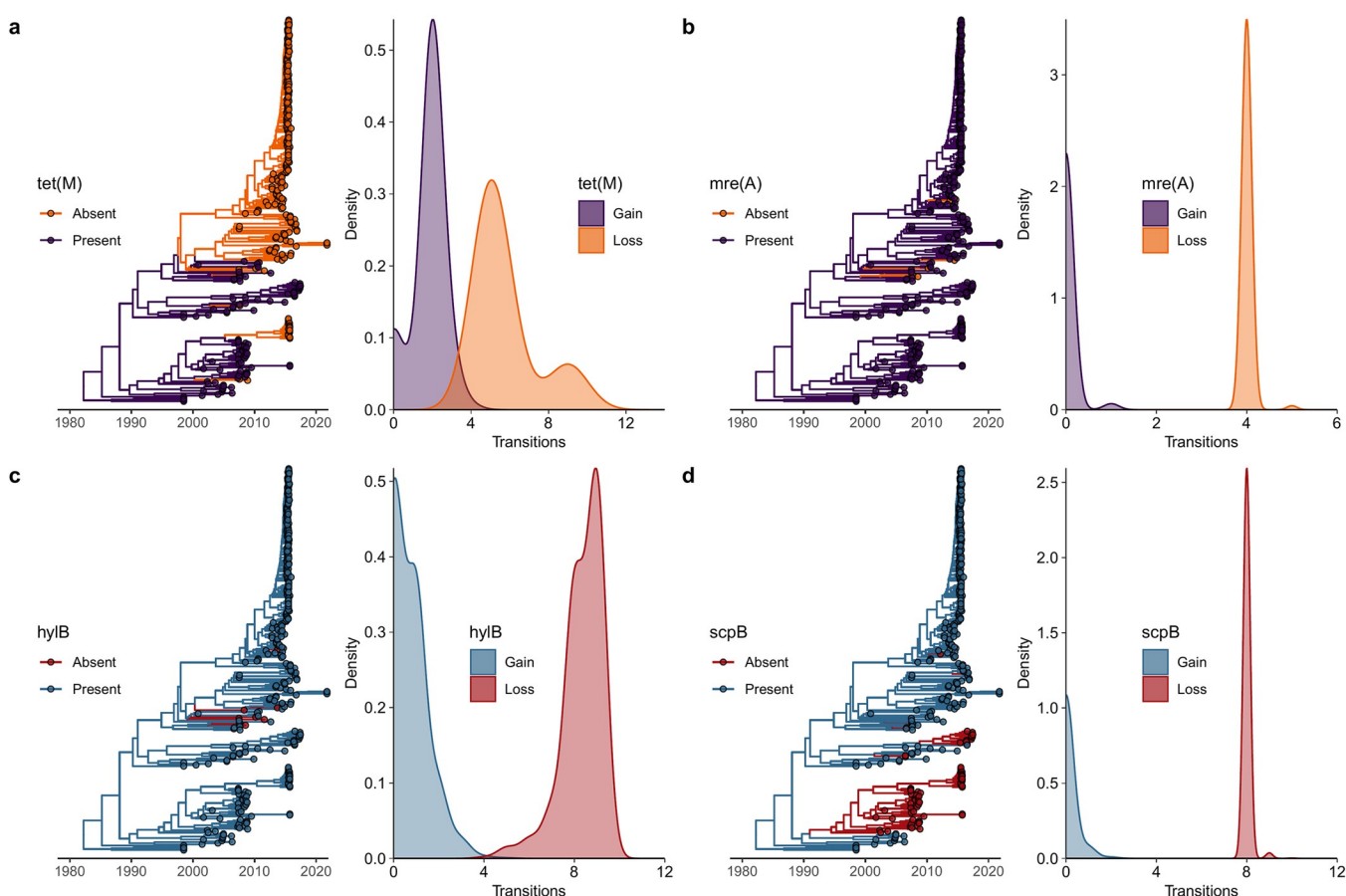

**Fig 4. Ancestral trait reconstructions as well as corresponding gain/loss of antimicrobial resistance and virulence factor genes.** Gene reconstructions are shown in the time-scaled maximum clade credibility (MCC) tree resulting from the Bayesian phylogenetic analysis and ancestral state reconstruction, with branches colored according to the most probable inferred gene state (absence or presence). The probability density of gene gains and losses are based on 1,000 trees sampled from the post burn-in posterior distribution. Displayed are **(a)** *tet*(M); **(b)** *mre*(A); **(c)** *hylB*; and **(d)** *scpB*.

each host type. Heterogeneous effort inherent to this dataset in sampling practices (point prevalence, event-based, and opportunistic) could affect these results. As such, these findings reflect the data currently available, and we cannot ascertain to what extent a different sampling effort would yield different results, which calls for further investigation. Nevertheless, the trend raises a possibility that ST283 may be maintained in humans residing in areas where sub-standard sanitation, hygiene and wastewater management facilitate repeat introductions into fish populations. GBS is recognized as a colonizer of the gastrointestinal tracts of healthy, asymptomatic humans. However, both the overall prevalence of ST283 carriage in the human population and the potential for human-to-human transmission of ST283 are unknown [13]. Although none of the 82 sampled food handlers and fishmongers in the 2015 Singapore outbreak carried ST283 [4], a study from northeast Thailand identified ST283 human fecal carriage in 5/184 (2.7%) samples, indicating human carriers could be involved in transmitting ST283 [20]. Human-associated GBS ST23 and ST7 have been isolated from marine mammals and from farm-raised crocodiles (ST23) and fish and amphibians (ST7) respectively, implicating anthropogenic pollution of the environment and surface waters in the switching between human and animal hosts [10, 21, 22]. Genome plasticity in GBS may facilitate adaptation to new ecological and host niches [23], and the propensity for GBS host switching has been

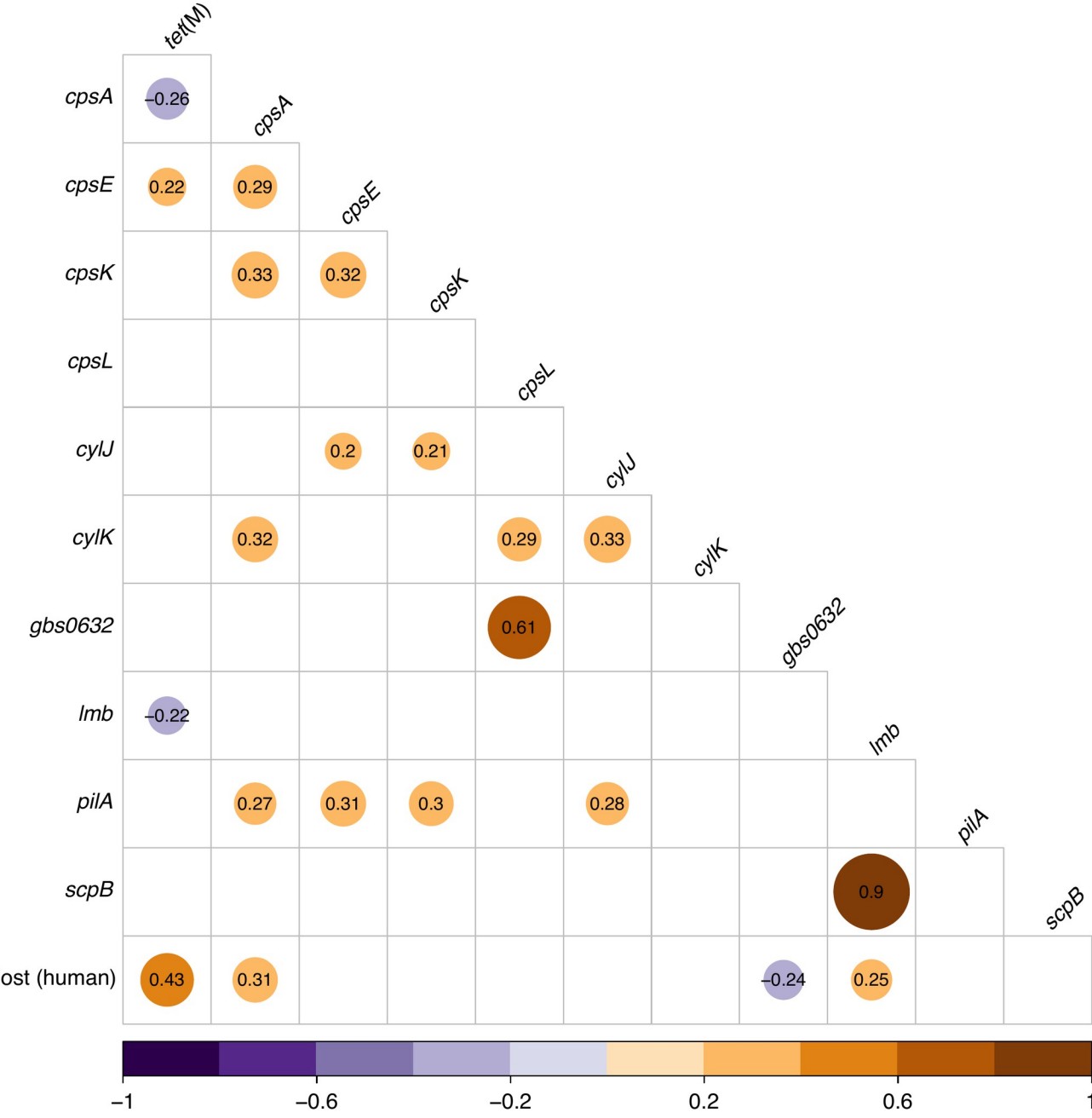

**Fig 5. Host and gene correlations across the evolutionary history of GBS ST283.** Gene-host and gene-gene partial correlations with a posterior median greater than 0.2 or less than– 0.2 (see Methods for further detail). Circle color represents the strength and direction of the correlation and circle size is proportional to the correlation magnitude. The host trait is coded as human = 1 and fish = -1.

recognized [23, 24]. Whether ST283 is maintained in fish or whether foodborne fish-to-human transmission is driven primarily by entry into the human food supply of fish during periodic outbreaks in these aquatic animals remains unknown. Maintenance in fish as a singular reservoir of ST283 could be expected to be associated with regular pulses of human foodborne infection in the absence of on-farm disease control measures. Such is the observed pattern for *Salmonella enterica* foodborne infection with continual human exposure through

the food chain associated with a *S. enterica* reservoir in laying hens [25]. In contrast, ST283 outbreaks in humans have been intermittent and stochastic, characterized by seasonal, short-duration episodes of case identification [4, 7]. Yet, if ST283 is maintained in fish, or if fish serve as an amplifying host subsequent to anthropogenic exposure, rapid growth in freshwater aquaculture in Asia beginning in the 1980s [19] could explain the observed increase in invasive GBS clinical disease first reported in Asia and highlight the significance of shifting dietary preferences in forewarning changes in foodborne disease incidence. Application of commercially available fish *Streptococcus agalactiae* vaccines may serve as a public health risk mitigation tool in limiting the potential role of fish as amplifying hosts and thus interrupting cyclical exchange of ST283 between humans and fish.

Human and fish exposure to a shared, as-yet undefined source must also be considered as GBS has a notoriously wide host range, the full extent of which for ST283 is undocumented [13]. Longitudinal surveillance of farmed fish and their rearing waters; establishing the prevalence of ST283 human carriage; and epidemiological investigation and source attribution of human outbreaks will be important in defining the cyclical movement of ST283 between humans and fish, the relative importance of each host as potential reservoirs, and the plausibility of a shared exposure source.

Antimicrobial resistance and virulence factor genes were lost more frequently than gained throughout the evolutionary history of ST283. An active rate of gene change in ST283 is consistent with GBS as a pathogen under evolutionary pressure, with recombination and mobile elements contributing to a shifting genome composition generating new, niche-adapted lineages [11, 23, 26]. Gene loss may facilitate rapid adaptation to the functional requirements of host suitability, for example, reducing the metabolic burden of gene expression, enhancing fitness and conferring a selective advantage in such niche-adapted lineages [23]. Our findings demonstrate a moderate rate of gain and loss for the tetracycline resistance gene *tet*(M) (Table 1). Previous work has shown that *tet*(M) is carried in GBS by the integrative and conjugative elements (ICE) Tn*916* and Tn*5801* [15, 16], and that the expansion of several tetracycline resistant human GBS clones and their global dissemination in the mid-twentieth century followed establishment of *tet*(M) through acquisition of these ICEs [15]. We identified a positive correlation between human host and *tet*(M) across the ST283 phylogeny and discrete trait analysis confirmed the previous finding of *tet*(M) gene presence at the ancestral root [16], suggesting it emerged from these tetracycline resistant human GBS clones and that the evolutionary path of ST283 has involved the repeated loss—and, in select instances, subsequent regain—of the *tet*(M) gene (Fig 4). We observe the apparent loss of *lmb* and *scpB* genes important in human invasive disease in the transition from human to fish (Figs 1C and 4D), which may be mediated by the ISSag2 insertion sequence element harboring these genes [27]. The *lmb* and *scpB* genes were positively correlated, and *lmb* was correlated with isolates originating from human hosts, pointing to a possible complementary role for these genes in invasive human infections. Loss of human-associated virulence factor genes in transitioning to fish hosts could explain the stochastic pattern of foodborne fish consumption-associated ST283 outbreaks: within a cyclical human-to-fish spillover and fish-to-human spillback context, the loss of GBS genes critical for human invasive disease may limit potential for spillback to result in human clinical illness [11].

Our analyses are subject to limitations. First, an extensive search of public repositories and literature enabled us to assemble the largest dataset of ST283 to date. Yet, despite this, the isolates identified may reflect neither the complete genetic diversity nor geographic distribution of ST283. South Asia—particularly India and Bangladesh—contributes substantial and growing freshwater fish production volume, yet no isolates were identified from this region, possibly reflecting a combination of differing strain presence patterns, exposure pathways, and

global disparities in typing and sequencing capacity. Efforts to type existing GBS strains and genetic characterization of GBS isolates associated with severe, invasive disease in adults may yield additional insights on GBS as a foodborne pathogen and associated exposure risk. Second, our dataset is characterized by sampling heterogeneity. While we account for this heterogeneous sampling effort in our analyses, the disparity in host, location, and sampling date could influence our findings. Third, whole genome sequences were primarily obtained as assemblies. The quality of genomes generated across differing sequencing and assembly methods could not be assessed. While genome quality could influence our findings, these genomes have been utilized in prior analyses. Finally, we did not have access to antimicrobial susceptibility testing for the isolates, limiting the ability to correlate presence of resistance genes with antimicrobial susceptibility profiles.

Recent work reported that despite nearly uniform carriage of *mre*(A) in a collection of ST283 isolates, none expressed resistance to macrolides [16]. In *Streptococcus agalactiae*, the *mre*(A) gene may serve a metabolic function, whereas when cloned into *E.coli*, *mre*(A) conferred macrolide resistance [28]. Additional study is needed to elucidate the phenotypic resistance profiles of ST283 and correlation with antimicrobial resistance genes. GBS phenotypic resistance to macrolides and fluoroquinolones vary regionally but remain concerning given the burden of invasive GBS infection globally [29]. Although the prevalence of multi-drug resistance gene carriage in ST283 is currently low, increasing use of medically important antimicrobials in the freshwater aquaculture industry [30] and in humans—particularly in low- and middle-income countries [31]—risks the generation of expanded resistance profiles.

Our findings demonstrate that ST283 holds the potential for geographic expansion, underscoring the need for enhanced surveillance across sectors, clinical awareness, and targeted risk mitigation and messaging. These findings highlight the importance of hygiene and sanitation, particularly in the context of the aquatic environment, wastewater management and along the food chain to limit transmission and mitigate the impact from this emerging pathogen.

## Methods

### Bacterial isolates

GBS ST283 whole genome sequences (*n* = 328) with corresponding sampling date and location were gathered through searches of public repositories (NCBI, ENA, and PubMLST; *n* = 3), literature review (PubMed; *n* = 26), research networks (*n* = 3) and from the Singapore *Streptococcus agalactiae* BioProject PRJNA293392 (*n* = 296). PubMed was searched for records in English through December 20, 2021 using search parameters: ("Group B *Streptococcus*" OR "*Streptococcus agalactiae*") AND (ST283 OR CC283). Genome sequences were excluded if they were not accompanied by a sampling date, sampling location, or host from which the sample was collected. Sequences from France were unavailable [8]. The 328 isolates were reportedly collected from humans (*n* = 251) and fish (*n* = 77) in eleven countries between 1998 and 2021 (S1 Data).

### Ethics statement

Ethics approval was obtained for three archived genome sequences from human clinical isolates from Hong Kong under the Joint Chinese University of Hong Kong–New Territories East Cluster Clinical Research Ethics Committee (CUHK-NTEC CREC) (ref. no.: 2018.509). The remaining 325 genome sequences were obtained from public repositories (accession numbers in S1 Data). The authors did not have access to individually identifiable data and no ethics committee approval for this study was sought.

## Whole genome sequence analysis

A total of 317 sequences were obtained as assemblies, and eleven as whole genome sequencing reads. The eleven whole genome sequence paired-end reads were assembled using SPAdes v.3.15.2 [32] with the "—careful" option. Quality of whole genome sequence reads was assessed using FastQC v.0.11.9 [33]. Quality assessment of assemblies generated in this study was assessed using QUAST v5.2.0 [34]. Mean Phred scores and assembly statistics (N50, L50) are reported in S1 Data.

## Phylogenetic analysis

GBS ST283 whole genome sequence assemblies were screened for antimicrobial resistance and virulence factor genes with Abricate [35] and AMRFinderPlus [36]. Assemblies were mapped to reference genome SG-M1 (accession CP012419.2) [37] and aligned using SKA v.1.0 [38]. The resulting 2.12 Mb whole genome alignment was analyzed using Gubbins [39] to identify and purge regions of recombination. Putative mobile genetic elements (MGE) were predicted and masked from the alignment. Single nucleotide polymorphisms (SNP) were called on this MGE-free and putatively non-recombinant alignment using SNP-sites [40]. A maximum likelihood (ML) phylogenetic tree was then inferred from the 1,214 SNP alignment using RAxML v8.2.12 [41] with a general time reversible (GTR) and gamma model of rate heterogeneity.

## Spatio-temporal and discrete trait analyses

To circumvent convergence issues with a joint inference approach and reduce computational burden, we used a multi-step process to generate spatial reconstructions of GBS ST283 transitions and to infer the gain and loss of antimicrobial resistance and virulence factor genes. In the first step we inferred a time-scaled phylogenetic tree from the ML tree using Markov chain Monte Carlo (MCMC) simulations over 10 million generations in the R package "BactDating" [42]. In the second step, we used this time-scaled phylogenetic tree as a fixed tree topology to perform both a discrete phylogeographic reconstruction and to analyze gene transition rates using the discrete diffusion model [43] implemented in the software package BEAST v1.10 [44]. In the discrete phylogeographic reconstruction, country transitions were estimated as Markov jump counts and reported along with both standard (Fig D in S1 Text) Bayes factor (BF) support values (ratio of posterior over prior odds and interpreted as a measure of the strength of evidence for the alternate hypothesis) and an adjusted (Fig 2) Bayes factor ($BF_{adj}$) supports [45] accounting for sample size disparity. Standard BF and $BF_{adj}$ values >20 were considered strong statistical support [46].

Host distribution in the dataset was associated with a heterogeneous sampling effort, with more isolates of human than fish origin (Fig 1). Sampling bias can introduce artifacts into discrete trait reconstructions [47], as confirmed by our $BF_{adj}$ support computation for the Markov jumps estimated between hosts (see the Results section). To account for this heterogeneous sampling orientation, we inferred host switching by working on downsampled time-scaled phylogenetic trees obtained by randomly sampling equal numbers of isolates originating from human and fish hosts in the tree tips. Three additional analyses were performed to assess the impact of sampling date and phylogenetic diversity associated with host type on host switching (Fig 3 and Methods in S1 Text) [48]. In each analysis, we generated 1,000 downsampled phylogenies, performing in each tree a maximum likelihood ancestral host estimation using an equal rates model implemented in the R package "ape" and counting host state transitions (human-to-fish and fish-to-human). The transition counts in each tree were taken as the distribution from which the median and interquartile range (IQR) were calculated.

### Across-trait host-gene and gene-gene correlations

To further investigate the dependencies between traits, we apply a recently developed phylogenetic multivariate probit model [49, 50], implemented in BEAST v1.10 [44], which can efficiently learn the correlation between discrete traits while adjusting for across-taxa covariation inherent to the phylogenetic tree. We report the across-trait partial correlations describing conditional dependencies between any two traits without confounding from other considered traits [49].

Details of the methods are provided in Methods in S1 Text.

## Supporting information

**S1 Text. File containing supplementary methods, tables, figures, and references.**
(PDF)

**S1 Data. File containing sampling date, location, host, genome characteristics and sequence accession numbers.**
(CSV)

## Author Contributions

**Conceptualization:** Daniel Schar, Simon Dellicour.

**Data curation:** Daniel Schar, Margaret Ip.

**Formal analysis:** Daniel Schar, Zhenyu Zhang, Joao Pires, Bram Vrancken, Marc A. Suchard, Philippe Lemey, Margaret Ip, Marius Gilbert, Thomas Van Boeckel, Simon Dellicour.

**Investigation:** Daniel Schar.

**Methodology:** Daniel Schar, Zhenyu Zhang, Joao Pires, Bram Vrancken, Simon Dellicour.

**Supervision:** Thomas Van Boeckel, Simon Dellicour.

**Visualization:** Daniel Schar, Zhenyu Zhang, Simon Dellicour.

**Writing – original draft:** Daniel Schar.

**Writing – review & editing:** Daniel Schar, Zhenyu Zhang, Joao Pires, Bram Vrancken, Marc A. Suchard, Philippe Lemey, Margaret Ip, Marius Gilbert, Thomas Van Boeckel, Simon Dellicour.

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
