## [Decision Letter · Decision Letter 0]

23 Aug 2023

PGPH-D-23-01077

Dispersal history and bidirectional human-fish host switching of invasive, hypervirulent *Streptococcus agalactiae* sequence type 283

Dear Dr. Schar,

Thank you for submitting your manuscript to PLOS Global Public Health. After careful consideration, we feel that it has merit but does not fully meet PLOS Global Public Health’s publication criteria as it currently stands. Therefore, we invite you to submit a revised version of the manuscript that addresses the points raised during the review process.

We look forward to receiving your revised manuscript.

Kind regards,

Ben Pascoe

Academic Editor

Journal Requirements:

2. Please provide separate figure files in .tif or .eps format only and remove any figures embedded in your manuscript file. Please also ensure all files are under our size limit of 10MB.

3. Some material included in your submission may be copyrighted. According to PLOS’s copyright policy, authors who use figures or other material (e.g., graphics, clipart, maps) from another author or copyright holder must demonstrate or obtain permission to publish this material under the Creative Commons Attribution 4.0 International (CC BY 4.0) License used by PLOS journals. Please closely review the details of PLOS’s copyright requirements here: PLOS Licenses and Copyright. If you need to request permissions from a copyright holder, you may use PLOS's Copyright Content Permission form.

Potential Copyright Issues:

Fig 1&2: please (a) provide a direct link to the base layer of the map (i.e., the country or region border shape) and ensure this is also included in the figure legend; and (b) provide a link to the terms of use / license information for the base layer image or shapefile. We cannot publish proprietary or copyrighted maps (e.g. Google Maps, Mapquest) and the terms of use for your map base layer must be compatible with our CC-BY 4.0 license. 

"

Additional Editor Comments (if provided):

Your manuscript has now been assessed by two reviewers and their feedback can be found below. Please consider the reviewers comments and revise your manuscript, particularly comments suggesting that you include a section on the quality and characteristics of the genomes used in the study.

Reviewers' comments:

Reviewer's Responses to Questions

**Comments to the Author**

1. Does this manuscript meet PLOS Global Public Health’s publication criteria? Is the manuscript technically sound, and do the data support the conclusions? The manuscript must describe methodologically and ethically rigorous research with conclusions that are appropriately drawn based on the data presented.

Reviewer #1: Yes

Reviewer #2: Yes

2. Has the statistical analysis been performed appropriately and rigorously?

Reviewer #1: Yes

Reviewer #2: Yes

3. Have the authors made all data underlying the findings in their manuscript fully available (please refer to the Data Availability Statement at the start of the manuscript PDF file)?

Reviewer #1: Yes

Reviewer #2: Yes

4. Is the manuscript presented in an intelligible fashion and written in standard English?

Reviewer #1: Yes

Reviewer #2: Yes

5. Review Comments to the Author

Reviewer #1: The authors have collated genomic information from a human and fish pathogenic clone of Streptococcus agalactiae ST283 to study its geographic spread and host dynamics. They have illustrated that the clone experienced increased dispersal to regions around the world since its emergence in the 1980’s. They also highlight that isolates have likely transmitted from humans to fish more frequently than from fish to humans. The data is somewhat biased since it is missing data from certain regions or contains more isolates from specific hosts or countries, however, this has been well documented in the text and statistical analyses have been carried out to account for and/or prove the effect of these biases. These findings highlight the potential for movement of virulent bacterial strains between aquaculture animals and susceptible human populations resulting in fish mortality and human infections. The authors advocate for further genomic surveillance and typing of Streptococcus agalactiae to further this work and help find mechanisms for reducing the spread of this pathogen. The manuscript is well written and most supplementary material and data has been included.

I have some comments and questions for the authors to consider.

Most important are:

1. I noticed that a lot of the genomes with variable AMR and virulence gene content in Figure 1 compared to the rest of the dataset were from one country, Laos. Are these isolates all from the same study and is it possible that the assemblies from these data were not as good resulting in missing data?

2. Can you include read quality and assembly statistics in the supplementary material?

Minor comments:

Line 58: “reduce impact from” -> “reduce the impact of”

Line 125: Consider expanding HPD for non-specialists.

Line 167: Are some of these genes intrinsic to Streptococcus agalactiae as a whole?

Figure 1: The names of AMR genes and virulence genes could be made bigger.

Figure S2: What does unrooted mean in this case?

Figure S4: I don't think this figure has been referenced in the text

Reviewer #2: Dear authors, attached are some suggestions for your consideration.

Line 71

The authors are encouraged to consider the following references if they find it appropriate.

Barkham, T., Tang, W. Y., Wang, Y. C., Sithithaworn, P., Kopolrat, K. Y., & Worasith, C. (2023). Human Fecal Carriage of Streptococcus agalactiae Sequence Type 283, Thailand. Emerging Infectious Diseases, 29(8), 1627.

Zhou, Y., Zhou, S., Peng, J., Min, L., Chen, Q., & Ke, J. (2023). Bacterial distribution and drug resistance in blood samples of children in Jiangxi Region, 2017–2021. Frontiers in Cellular and Infection Microbiology, 13, 1163312.

Sirimanapong, W., Phước, N. N., Crestani, C., Chen, S., & Zadoks, R. N. (2023). Geographical, Temporal and Host-Species Distribution of Potentially Human-Pathogenic Group B Streptococcus in Aquaculture Species in Southeast Asia. Pathogens, 12(4), 525.

Luangraj, M., Hiestand, J., Rasphone, O., Chen, S. L., Davong, V., Barkham, T., ... & Keoluangkhot, V. (2022). Invasive Streptococcus agalactiae ST283 infection after fish consumption in two sisters, Lao PDR. Wellcome open research, 7.

Wang, J., Zhang, Y., Lin, M., Bao, J., Wang, G., Dong, R., ... & Pan, X. (2023). Maternal colonization with group B Streptococcus and antibiotic resistance in China: systematic review and meta-analyses. Annals of clinical microbiology and antimicrobials, 22(1), 5.

Line 153

It is possible to consider incorporating the availability and impact of the effort and the ability to obtain human or fish sequences in these results?

Line 215 Figure 1 and “Supplementary Figure S2a”.

The suggestion is to clarify and standardize the names of supplementary information and supplementary figures. It is not easy to track the information as it is currently named.

The suggestion is to systematically list and number the information so that the reader can quickly identify it.

Line 341

It is suggested to include more detailed information about the genomes, sequencing methods, sequence quality parameters, and criteria for inclusion and exclusion of genomes in the analysis. Certain aspects, such as the reference sequence and whether the available genomes are fully assembled or assembled from published partial sequences, sequencing methods should be included in the Materials and Methods section. It is suggested that certain elements of the materials and methods, as described in 'Click here to access/download Supporting Information: Schar_D_et_al_PLOSGPH_Supplementary_Information.pdf', should be included in the document to achieve a comprehensive understanding of it.

Line 359

Figure 1: The suggestion is that the table displaying the presence or absence of antimicrobial resistance genes (blue) and virulence factor genes (red) for each isolate in the study (Table "D") be considered. The suggestion is to condense the information in the figure/table. Specifically, for the table of resistance genes, all sequences with the same pattern can be removed, retaining only a few representative ones. he letters are too small.

The suggestion is to thoroughly discuss the effects of sample availability on the statement regarding the transmission of bacteria from humans to fish or from fish to humans. Additionally, it is recommended to provide a basis for the loss of virulence and resistance genes as an evolutionary strategy.

6. PLOS authors have the option to publish the peer review history of their article (what does this mean?). If published, this will include your full peer review and any attached files.

**Do you want your identity to be public for this peer review?** For information about this choice, including consent withdrawal, please see our Privacy Policy.

Reviewer #1: No

Reviewer #2: No

---

## [Editor Report · Decision Letter 1]

27 Sep 2023

Dispersal history and bidirectional human-fish host switching of invasive, hypervirulent *Streptococcus agalactiae* sequence type 283

PGPH-D-23-01077R1

Dear Dr. Schar,

We are pleased to inform you that your manuscript 'Dispersal history and bidirectional human-fish host switching of invasive, hypervirulent *Streptococcus agalactiae* sequence type 283' has been provisionally accepted for publication in PLOS Global Public Health.

Best regards,

Ben Pascoe

Academic Editor

Thanks you for considering all reviewer feedback and providing a point by point response. Having taken on board the reviewer comments and incorporated additional material, including details on sequencing and genome curation, I can now recommend acceptance of your manuscript.